# Value of Serum Sirtuin-1 (SIRT1) Levels and SIRT1 Gene Variants in Periodontitis Patients

**DOI:** 10.3390/medicina58050653

**Published:** 2022-05-12

**Authors:** Albertas Kriaučiūnas, Rasa Liutkevičienė, Greta Gedvilaitė, Kristė Kaikarytė, Alvita Vilkevičiūtė, Darius Gleiznys, Ingrida Pacauskienė, Gediminas Žekonis

**Affiliations:** 1Department of Prosthodontics, Lithuanian University of Health Sciences, Sukilėlių Str. 51, LT-50106 Kaunas, Lithuania; darius.gleiznys@lsmuni.lt (D.G.); gediminas.zekonis@lsmu.lt (G.Ž.); 2Institute of Neuroscience, Lithuanian University of Health Sciences, Eivenių Str. 2, LT-50009 Kaunas, Lithuania; rasa.liutkeviciene@lsmu.lt (R.L.); greta.gedvilaite@lsmu.lt (G.G.); kriste.kaikaryte@lsmuni.lt (K.K.); alvita.vilkeviciute@lsmu.lt (A.V.); 3Department of Dental and Oral Pathology, Lithuanian University of Health Sciences, Eivenių Str. 2, LT-50009 Kaunas, Lithuania; ingridamarija.pacauskiene@lsmuni.lt

**Keywords:** periodontitis, *SIRT1*, rs3758391, rs3818292, rs7895833, *SIRT1* levels

## Abstract

*Background and Objectives*: Periodontitis is a multifactorial inflammatory disease associated with biofilm dysbiosis and is defined by progressive periodontium destruction. Genes largely regulate this entire process. *SIRT*s are a group of histone deacetylases (HDACs) intimately involved in cell metabolism and are responsible for altering and regulating numerous cell functions. Understanding *SIRT*s and their functions in periodontitis may be useful for therapeutic treatment strategies in the future. The aim of our study was to investigate the associations amid *SIRT1* single-gene nucleotide polymorphisms (rs3818292, rs3758391, and rs7895833) and *SIRT1* serum levels for patients affected by periodontitis in the Caucasian population. *Materials and Methods*: The study included 201 patients affected by periodontitis and 500 healthy controls. DNA extraction from peripheral leukocytes was carried out using commercial kits. The real-time PCR method was selected for the determination of the genotype of the periodontitis patients and the control group. The ELISA method was used to measure the *SIRT1* concentration. A statistical data analysis was performed using “BM SPSS Statistics 27.0” software. *Results*: The *SIRT1* rs3818292 AG genotype was associated with a 2-fold and 1.9-fold increase in the development of periodontitis under the codominant and overdominant models (OR = 1.959; CI = 1.239–3.098; *p* = 0.004; and OR = 1.944; CI = 1.230–3.073; *p* = 0.004, respectively). The serum *SIRT1* levels were not statistically significantly different between subjects in the periodontitis and control groups (0.984 (5.159) ng/mL vs. 0.514 (7.705) ng/mL, *p* = 0.792). *Conclusions*: in our study, the genotypes and alleles of *SIRT1* rs3818292, rs3758391, and rs7895833 statistically significantly differed between the periodontitis and control groups, exclusively in the male population and subjects older than 60 years.

## 1. Introduction

Periodontitis is a complex inflammatory disease, which is interconnected with a dysbiotic biofilm and is characterized by progressive destruction of the periodontium [1]. It is approximated that 538 million people globally are affected by a severe form of periodontitis [2]. Periodontal disease etiology is multifactorial. The host inflammatory and immune response is triggered by the subgingival dental biofilm. This eventually leads to permanent destruction of the periodontium (i.e., alveolar bone and periodontal ligament) in a susceptible host [3].

In the patient’s mouth, the microbiota, immune system, and lifestyle habits interact with each other. This can lead to multiple changes, and the physiology must adapt to keep the host healthy. This whole interplay is regulated mainly by genes [4]. *SIRTs* are a group of histone deacetylases (HDACs), which are divided into seven different members (from *SIRT1* to *SIRT7*) [5]. These proteins are expressed in normal tissues and have an impact on numerous biological processes [6]. Sirtuins, a group of nicotinic adenine dinucleotide (NAD+)-dependent enzymes, are closely related to the metabolism of the cell and are responsible for altering and regulating numerous cellular functions, such as DNA repair, inflammatory responses, apoptosis, aging, or cell cycle [7,8]. Although they belong to the same family, the location of each gene is different. The nucleus locates *SIRTs 1*, *6*, and *7*, while *SIRT2* is found in the cytosol, and the remaining *SIRTs 3*, *4*, and *5* are detected in the mitochondria [9]. One of the most studied members of the family is *SIRT1*, which enacts in metabolic health by deacetylation of numerous target proteins in different tissues (heart, muscle, endothelium, liver etc.) and is well known to affect cancer by its suppression or promotion, depending on the cell type or its content [10]. It is also a gene found in all living organisms and is known as the longevity gene [11]. While focusing on various oral diseases and pathologies, the role of some members of the sirtuin gene family remains unclear to date, such as *SIRT2*, whose part in oral cancer pathogenesis has not yet been looked into. Recently, a correlation between oral malignancies and the *SIRT1*, *SIRT3*, and *SIRT7* genes and their expressions has been reported [9]. In periodontal diseases, although not directly, some sirtuins (*SIRT4*- and *SIRT3*-families) are studied, and there tend to be associations with various pathologies, such as diabetes melitus [12]. Some members of the *SIRT* family, especially *SIRT6*, tend to have a therapeutic effect on periapical lesions and their treatment by suppressing CCL2 synthesis, which is also associated with regulatory activities in cellular metabolism [13].

Therefore, understanding *SIRT* and its functions in oral diseases, such as periodontitis, may be useful for therapeutic strategies in the future [14,15]. *SIRT1* had a beneficial effect in reducing vascular senescence in the endothelial cell culture model [16]. An increase in levels of *SIRT1* is shown to prevent the progression of periodontal disease in an animal study [17]. Additionally, Caribe et al. have reported that *SIRT1* levels tend to increase after periodontal treatment [18,19]. However, a limited number of studies show the association between elevated serum *SIRT1* and periodontitis. Therefore, understanding *SIRT* and its functions in oral diseases, such as periodontitis, may be useful for therapeutic treatment strategies in the future.

The aim of our study was to examine the links connecting *SIRT1* single-gene nucleotide polymorphisms (rs3818292, rs3758391, and rs7895833) and *SIRT1* serum levels in patients with periodontitis in the Caucasian population.

## 2. Materials and Methods

The case–control study was conducted at the Department of Prosthodontics, Lithuanian University of Health Sciences Hospital, and the Neuroscience Institute of the Lithuanian University of Health Sciences.

### 2.1. Ethics Statement

The Lithuanian University of Health Sciences Ethics Committee has approved the study for Biomedical Research (No. BE-2-20). All subjects gave written informed consent in accordance with the Declaration of Helsinki.

### 2.2. Control and Patients with Periodontitis Group Formation

The study consisted of 201 patients with periodontitis (*n* = 201) and 500 healthy control subjects (*n* = 500).

All the patients selected for the study met the following criteria: (i) over 18 years of age, (ii) had more than 30% of their mouth affected by periodontitis, leading to a diagnosis of generalized periodontal disease, (iii) consented to intraoral and radiographic examination to determine the extent of periodontitis.

Patients were excluded from the study if they met any of the following statements: (i) had undergone chemotherapy, (ii) tooth loss was not due to periodontal disease but to other causes (patients’ previous dental records indicating trauma and tooth extractions due to treatment of jaw fractures), (iii) no radiological examination with orthopantomogram was performed, (iv) the mouth was completely edentulous, (v) systemic diseases, e.g., diabetes mellitus, malignant tumors, systemic connective tissue diseases, (vi) chronic infectious/and non-infectious diseases or conditions after organ or tissue transplantation, (vii) diseases of the cardiovascular system; (viii) regular smokers and alcohol abusers.

### 2.3. Odontological Examination

The study participants underwent intraoral and radiographic examinations at their first visit to the prosthodontist. Periodontal disease was diagnosed according to the consensus report of Working Group 2 of the World Workshop on the Classification of Periodontal and Peri-Implant Diseases and Conditions 2017 [20]. The study included patients with all three different stages of periodontitis (I, II, and III). The extent of periodontitis was >30% of the total affected teeth in the subject’s mouth, in all the patients. The periodontitis-affected patients underwent radiographic examination to determine if bone loss was present and to evaluate it (orthopantomogram, dental radiographs and proximal digital radiographs were examined).

### 2.4. DNA Extraction, Genotyping, and Enzyme-Linked Immunosorbent Assay

DNA samples were extracted from peripheral venous blood using the DNA salting-out method. Genotyping of all three SNPs was performed using TaqMan^®^ genotyping assays (Applied Biosystems, Foster City, CA, USA) and *SIRT1* (rs3818292, rs3758391, and rs7895833) according to the manufacturer’s instructions using real-time polymerase chain reaction (PCR). Serum *SIRT1* levels were determined in 500 control subjects and 201 patients with periodontitis. Serum *SIRT1* levels in patients were determined using the commercial enzyme-linked immunosorbent assay (ELISA) kit for human *SIRT1* (Human SIRT1 ELISA Kit, Abcam, Cambridge, United Kingdom) according to the manufacturer’s instructions, and optical density was measured immediately at a wavelength of 450 nm, using a microplate reader (Multiskan FC microplate photometer, Thermo Scientific, Waltham, MA, USA). The *SIRT1* level was calculated using the standard curve; sensitivity range of the standard curve was 0.63–40 ng/mL, sensitivity 132 pg/mL [21].

### 2.5. Statistical Analysis

SPSS/W 27.0 software (Statistical Package for the Social Sciences for Windows, Inc., Chicago, IL, USA) was used for the statistical analysis. Absolute numbers with percentages were used for data expression. Percentages were used for genotype frequencies expressions. A Hardy–Weinberg analysis was performed to compare the observed and expected frequencies of *SIRT1* (rs3818292, rs3758391, and rs7895833) with the χ^2^ test in all the groups. The χ^2^ test was used for the distribution comparison of *SIRT1* (rs3818292, rs3758391, and rs7895833) in the periodontitis and control groups. The estimation of the influence of genotypes on the development of periodontitis was performed by the binomial logistic regression analysis. Odds ratios and 95% confidence intervals were presented. The best genetic model selection was based on the Akaike Information Criterion (AIC); consequently, the lowest AIC values represented the best genetic models.

## 3. Results

There was no deviation of the genotypes of the tested SNPs from the Hardy–Weinberg equilibrium (HWE) (*p* > 0.05). The study group included 201 patients with periodontitis; 85 (42.73%) were men and 116 (57.7%) were women. The median age was 70.0 years (IQR = 16). The control group consisted of 500 subjects; 250 were men and 250 women; the median age was 66.00 years (IQR = 15). The age and gender did not statistically significantly differ between the groups (*p* = 0.082, *p* = 0.065, respectively) (Table 1).

The genotypes and allele distributions of *SIRT1* rs3818292 and rs7895833 were statistically significantly different between the periodontitis and the control group. The analysis showed that the *SIRT1* rs3818292 AA genotype was less frequent in the periodontitis group than in the control group (80.6% vs. 89.2%, *p* = 0.002), while the AG genotype was more frequent in the periodontitis group than in the control group (18.4% vs. 10.4%, *p* = 0.004). The G allele was more frequent in the periodontitis group than in the control group (10.2% vs. 5.4%, *p* = 0.001). The *SIRT1* rs7895833 AA genotype was less frequent in the periodontitis group than in the control group (66.2% vs. 76.6%, *p* = 0.005), while AG was more frequent in the periodontitis group than in the control group (32.3% vs. 22.2% *p* = 0.005). The G allele was more frequent in the periodontitis group than in the control group (17.7% vs. 12.3%, *p* = 0.009) (Table 2).

Further binary logistic regression analysis was conducted to evaluate the effects of these SNPs on periodontitis in males. The analysis revealed that the *SIRT1* rs3818292 AG genotype was associated with 2-fold and 1.9-fold increased odds of periodontitis development under the codominant and overdominant models (OR = 1.959; CI = 1.239–3.098; *p* = 0.004; and OR = 1.944; CI = 1.230–3.073; *p* = 0.004, respectively). The AG + GG genotypes were also associated with 2-fold increased odds of periodontitis development under the dominant model (OR = 1.988; CI = 1.269–3.116; *p* = 0.003). Each G allele increased the odds of periodontitis development by 1.9-fold in the additive model (OR = 1.905; CI = 1.249–2.906; *p* = 0.003). The *SIRT1* rs7895833 AG genotype was associated with 1.7-fold increased odds of periodontitis development under the codominant and overdominant models (OR = 1.686; CI = 1.172–2.427; *p* = 0.005; and OR = 1.675; CI = 1.165–2.408; *p* = 0.005, respectively). The AG + GG genotypes were associated with 1.7-fold increased odds of periodontitis development under the dominant model (OR = 1.674; CI = 1.170–2.394; *p* = 0.005). Each G allele increased the odds of periodontitis development by 1.6-fold in the additive model (OR = 1.572; CI = 1.129–2.188; *p* = 0.007) (Table 3).

The genotype and allele distributions of *SIRT1* rs3818292 and rs7895833 were statistically significantly different between the periodontitis and control groups when distinguished between gender. The analysis revealed that the *SIRT1* rs3818292 AA genotype was less frequent in the male periodontitis group than in the control group (77.6% vs. 88.8%, *p* = 0.010), while the AG genotype was more frequent in the periodontitis group than in the control group (22.4% vs. 10.8%, *p* = 0.008). The G allele was more frequent in the male periodontitis group than in the control group (11.2% vs. 5.8%, *p* = 0.019). In addition, the *SIRT1* rs7895833 AA genotype was less frequent in the periodontitis group than in the control group (60.0% vs. 75.2%, *p* = 0.007), while the AG genotype was more frequent in the periodontitis group than in the control group (37.6% vs. 23.6% *p* = 0.012). The G allele was more frequent in the periodontitis group than in the control group (21.2% vs. 13.0%, *p* = 0.010) (Table 4). In addition, the analysis revealed that the *SIRT1* rs3818292 G allele was more frequent in the female periodontitis group than in the control group (9.5% vs. 5.4%, *p* = 0.001) (Table 5).

In addition, we performed a binary logistic regression analysis to evaluate the effects of these SNPs on periodontitis in males. The analysis revealed that the *SIRT1* rs3818292 AG genotype was associated with 2.4-fold increased odds of periodontitis development under the codominant and overdominant models (OR = 2.367; CI = 1.238–4.525; *p* = 0.009 and OR = 2.378; CI = 1.244–4.545; *p* = 0.009, respectively). In addition, the AG + GG genotypes were associated with 2.3-fold increased odds of periodontitis development under the dominant model (OR = 2.282; CI = 1.199–4.347; *p* = 0.012). Each G allele increased the odds of periodontitis development by 2.1-fold in the additive model (OR = 2.110; CI = 1.129–3.945; *p* = 0.019). The *SIRT1* rs7895833 AG genotype was associated with 2-fold increased odds of periodontitis under the codominant and overdominant models (OR = 1.999; CI = 1.177–3.397; *p* = 0.01 and OR = 1.955; CI = 1.154–3.311; *p* = 0.013, respectively). The *SIRT1* rs7895833 AG + GG genotypes were also associated with 2-fold increased odds of developing periodontitis under the dominant model (OR = 2.022; CI = 1.201–3.401; *p* = 0.008). Each G allele increased the odds of periodontitis development by 1.9-fold in the additive model (OR = 1.888; CI = 1.173–3.038; *p* = 0.009) (Table 6). Therefore, no statistically significant associations were found in the women’s periodontitis and control groups (data not shown).

The genotypes and allele distributions of *SIRT1* rs3818292 and rs7895833 were statistically significantly different between the periodontitis group and the control group of patients older than 60 years. The analysis showed that the *SIRT1* rs3818292 AA genotype was less frequent in the periodontitis group than in the control group (78.1% vs. 89.4%, *p* = 0.001), while the AG genotype was more frequent in patients over 60 years old than in the control group (20.5% vs. 10.6%, *p* = 0.003). The G allele was more frequent in the periodontitis group than in the control group (11.6% vs. 5.2%, *p* = 0.001). The genotype of *SIRT*1 rs7895833 AA was less frequent in the periodontitis group than in the control group (64.2% vs. 76.1%, *p* = 0.006), while the genotype AG was more frequent in the periodontitis group than in the control group (33.8% vs. 22.8% *p* = 0.001). The G allele was more frequent in the periodontitis group than in the control group (18.9% vs. 12.5%, *p* = 0.008) (Table 7). No statistically significant associations were found between the periodontitis and control groups within the 60-year-old or younger population (data not shown).

Moreover, a binary logistic regression analysis was also performed in the group of patients older than 60 years to evaluate the impact of these SNPs on periodontitis. The analysis revealed that the *SIRT1* rs3818292 AG genotype was associated with 2.2-fold increased odds of periodontitis development under the codominant and overdominant models in the over-60-year-old group (OR = 2.216; CI = 1.322–3.714; *p* = 0.003; and OR = 2.179; CI = 1.301–3.650; *p* = 0.003, respectively). In addition, the AG + GG genotypes were associated with 2.4-fold increased odds of periodontitis development according to the dominant model (OR = 2.359; CI = 1.418–3.925; *p* = 0.001). Each G allele increased the odds of periodontitis development by 2.4-fold in the additive model (OR = 2.408; CI = 1.474–3.933; *p* = 0.001). The *SIRT1* rs7895833 AG genotype was associated with a 1.8-fold increase in the odds of periodontitis development in the codominant model and a 1.7-fold increase in the overdominant model (OR = 1.753; CI = 1.154–2.661; *p* = 0.008; and OR = 1.724; CI = 1.138–2.614; *p* = 0.01, respectively). The AG + GG genotypes were also associated with 1.8-fold increased odds of developing periodontitis under the dominant model (OR =1.771; CI = 1.176–2.669; *p* = 0.006). Each G allele increased the odds of periodontitis development by 1.7-fold in the additive model (OR = 1.688; CI = 1.157–2.463; *p* = 0.007) (Table 8).

Figure 1 shows the *SIRT1* serum levels in periodontitis patients and control group subjects. An evaluation of serum *SIRT1* levels was performed in seven periodontitis-affected patients and five control group subjects. We found that *SIRT1* serum levels were not statistically significantly different between the periodontitis patients and control group subjects (0.984 (5159) ng/mL vs. 0.514 (7705) ng/mL, *p* = 0.792) (Figure 1).

## 4. Discussion

Several factors and mechanisms must be considered when studying the relationship between periodontitis and sirtuin activity. In research carried out by Caribe et al., *SIRT1* was linked with protection against inflammation when two groups of patients (forty periodontal patients and thirty-eight healthy individuals) were compared before and after periodontal treatment; *SIRT1* levels tend to increase after periodontal treatment [18]. In another study, researchers Kude et al. investigated periapical periodontitis. The results showed that *SIRT1* might affect angiogenesis in periapical granulomas, and the mechanism was based on the activation of endothelial cell proliferation, along with VE-cadherin and VEGF expression [22]. It is important to note that in mammalian cells, *SIRT1* works as a regulator for the release suppression of inflammatory mediators. A study by Qu et al. showed that the activation of *SIRT1* significantly impaired and suppressed the expression of MMP-13 by focusing on NF-κB p65 [23]. Periodontitis has also been associated with numerous other diseases, including coronary artery disease. Caribe et al. conducted a study showing that periodontal disease treatment decreased the mannose-binding lectin concentration (a protein of the immune system that tends to bind to pathogens in periodontal disease) and increased the serum *SIRT1* concentration in the group of patients who were and were not affected by coronary artery disease [19].

Nevertheless, cell regeneration is another important aspect of analyzing periodontal treatment options. The research results of Lee et al. covered *SIRT1* activation by resveratrol in various cells; periodontal ligament cells, cementum, and osteoblasts showed an increased mineralized nodules formation and the over-expression of mRNAs [24]. It should also be noted that the overexpression of *SIRT1* promotes periodontal ligament cell differentiation into osteoblast-like cells and affects the said cells’ mineralization, while periodontal cell differentiation is blocked by the suppression of *SIRT1* [24]. In a study by Zhang et al., rs3758391/CC was found to be more prevalent in comparison to rs3758391/CT and rs3758391/TT in subjects who were older than 60 years (odds ratio = 3.042, *p* = 0.027) [25], although in our research, there were no statistically significant results related to periodontal inflammation for the same polymorphism and age group. Furthermore, Yue et al. investigated the associations between *SIRT1* gene polymorphisms and diabetic kidney disease, where rs3818292 patients with the GG genotype in the rs3818292 locus were 0.23-fold and 0.21-fold higher than for AA and for AA + AG genotypes, respectively, which in our case, the AG + GG genotypes were associated with 2-fold increased odds of periodontitis development in the dominant model (OR = 1.988; CI = 1.269–3.116; *p* = 0.003). In the work of Hou et al., the AA genotype of rs7895833 was associated with a significantly decreased risk of CRS1 (OR = 0.49), whereas in our study, the genotype AA was less frequent in the periodontitis patients’ group, in comparison to the control group (60.0% vs. 75.2%, *p* = 0.007) in the male population. The following study also proved that the two *SIRT1* rs1467568 and rs7895833 polymorphisms had an impact on the reduced risk of developing CRS1 in the Chinese population [26]. In the work of Vaiciulis et al., which focused on the study of laryngeals squamous cell carcinoma (LSCC) development probability, carriers of the *SIRT1* rs3758391 T/T genotype had a statistically significant increased probability of developing advanced-stage LSCC, according to the codominant and recessive genetic model (OR = 2.387; 95% CI = 1.091–5.222; *p* = 0.029 and OR = 2.287; 95% CI = 1.070–4.888; *p* = 0.033, respectively) [27]. In contrast, in our study, we found no statistically significant differences in the development of periodontitis in the male population in our study.

The *SIRT2* family was also shown to have an impact on periodontal disease cases. Kluknavska et al. concluded that *SIRT2* is increased in aggressive and chronic periodontitis, compared to healthy individuals [28]. On the other hand, *SIRT3* plays a functional role in age-related periodontal diseases and their underlying mechanisms. As Chen et al. found, a decrease in *SIRT3* abundance affects age-related periodontal disease by exacerbating oxidative stress and promoting mitochondrial dysfunction [29]. In addition, *SIRT3* and *SIRT4* play an important role in analyzing blood glucose levels in patients suffering from diabetes melitus. This is important for early diagnosis and for people who do not respond to other drugs [12]. It is important to note that *SIRT3* and its inhibitors are a promising new avenue to explore the development of therapies for other oral diseases, such as head and neck cancer, by inhibiting cell proliferation and promoting apoptosis [30]. In the work of Yang et al., which focused on the role of *SIRT5* in apical periodontitis, the results showed that *SIRT5* expression decreased, oxidative stress increased, and apoptosis was enhanced in bone tissue cells [14]. This led the researchers to conclude that increasing *SIRT5* may potentially be a therapeutic treatment for apical periodontitis [14]. As for the activity of *SIRT6* in the pathogenesis of periapical lesions, a study by Kok et al. showed that *SIRT6* has an effect on the suppression of glycolysis enhanced by hypoxia and can inhibit apoptosis induced by hypoxia or treatment with lactate [31]. This indicates that *SIRT6* plays a role as a negative regulator of inflammation. It may also positively alleviate periapical lesions by suppressing osteoblastic glycolysis and apoptosis [31]. Researchers Lee et al. also investigated the ability of *SIRT6* to suppress the synthesis of CCL2 (chemokine ligand 2), which may lead to a therapeutic effect on periapical lesions [13]. As noted by Huang et al., *SIRT6* also played a negative role in the differentiation and mineralization of cementoblasts, which was important because of the similar composition of cementum and its importance in homeostasis during periodontal treatment [32]. Although no recent data on *SIRT7* and periodontal disease could be found at the time of this review, downregulation of *SIRT7* is observed in carcinomas. A study by Li et al. showed that *SIRT7* is downregulated in OSCC cell lines and human OSCC/OSCC tissues with lymph node metastases, which functions by preventing SMAD4 deacetylation [33]. Further long-term studies are needed to investigate and evaluate the changes and their impact on the above biomarkers, in terms of prognosis and further progression of periodontal disease.

Some limitations of this study must be acknowledged. The sample size was relatively small in our study, and further research with a larger sample is needed.

## 5. Conclusions

*SIRT1* rs3818292 and rs7895833 might be associated with decreased odds of PD development exclusively in the male population, in subjects older than 60 years. Serum *SIRT1* levels were not statistically significantly different between subjects in the periodontitis and control groups.

## Figures and Tables

**Figure 1 medicina-58-00653-f001:**
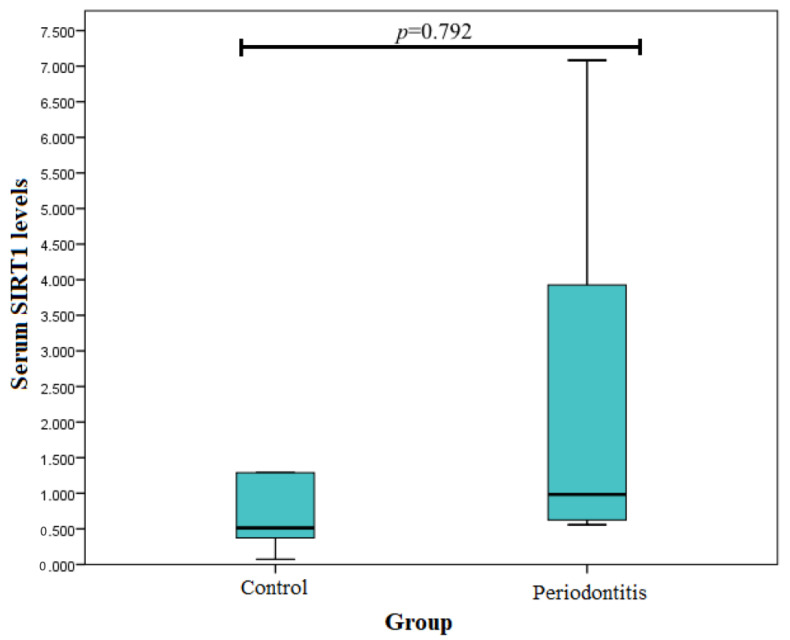
*SIRT1* serum levels in periodontitis and control group subjects.

**Table 1 medicina-58-00653-t001:** Demographic characteristics of the study population.

Characteristic	Group		*p*-Value
	Periodontitis *n* = 201	Control Group *n* = 500	
Males, *n* (%)	85 (42.73%)	250 (50%)	0.065
Females, *n* (%)	116 (57.7%)	250 (50%)	
Age, median (IQR)	70.0 (16)	66.00 (15)	0.082

*p*-value—significance level.

**Table 2 medicina-58-00653-t002:** Genotype and allele distribution of *SIRT1* rs3818292, rs3758391, and rs7895833 in periodontitis and control groups.

Genotype/Allele	Periodontitis *n* = 201 *n* (%)	Control Group *n* = 500 *n* (%)	HWE *p*-Value	*p*-Value
**rs3818292**				
AA	162 (80.6) ^1^	446 (89.2) ^1^	0.715	**0.009**
AG	37 (18.4) ^2^	52 (10.4) ^2^		
GG	2 (1.0)	2 (0.4)		
Total	201 (100.0)	500 (100.0)		
A	361 (89.8)	976 (94.6)		**0.001**
G	41 (10.2)	56 (5.4)		
**rs3758391**				
CC	103 (51.2)	288 (57.6)	0.179	0.272
CT	86 (42.8)	190 (38.0)		
TT	12 (6.0)	22 (4.4)		
Total	201 (100.0)	500 (100.0)		
C	292 (72.6)	766 (76.6)		0.119
T	110 (27.4)	234 (23.4)		
**rs7895833**				
AA	133 (66.2) ^3^	383 (76.6) ^3^	0.517	**0.018**
AG	65 (32.3) ^4^	111 (22.2) ^4^		
GG	3 (1.5)	6 (1.2)		
Total	201 (100.0)	500 (100.0)		
A	331 (82.3)	877 (87.7)		**0.009**
G	71 (17.7)	123 (12.3)		

*p*-value—significance level; ^1^ (AA vs. AG + GG) *p* = 0.002; ^2^ (AG vs. AA + GG) *p* = 0.004; ^3^ (AA vs. AG + GG) *p* = 0.005; ^4^ (AG vs. AA + GG) *p* = 0.005.

**Table 3 medicina-58-00653-t003:** Binomial logistic regression analysis of periodontitis and control groups.

Model	Genotype/Allele	OR (95% CI)	*p*-Value	AIC
**Periodontitis**				
***SIRT1* rs3818292**				
Co-dominant	AG vs. AA GG vs. AA	1.959 (1.239–3.098) 2.753 (0.385–19.706)	**0.004** 0.313	835.292
Dominant	AG + GG vs. AA	1.988 (1.269–3.116)	**0.0** **03**	833.402
Recessive	GG vs. AG + AA	2.503 (0.350–17.888)	0.361	841.271
Overdominant	AG vs. AA + GG	1.944 (1.230–3.073)	**0.004**	834.269
Additive	G	1.905 (1.249–2906)	**0.003**	833.384
***SIRT1* rs3758391**				
Co-dominant	CT vs. CC TT vs. CC	1.266 (0.901–1.778) 1.525 (0.729–3.192)	0.174 0.263	841.503
Dominant	CT + TT vs. CC	1.293 (0.931–1.795)	0.126	839.739
Recessive	TT vs. CT + CC	1.380 (0.669–2.843)	0.383	841.343
Overdominant	CT vs. CC + TT	1.220 (0.875–1.702)	0.241	840.711
Additive	T	1.252 (0.952–1.646)	0.108	839.515
***SIRT1* rs7895833**				
Co-dominant	AG vs. AA GG vs. AA	1.686 (1.172–2.427) 1.440 (0.355–5.838)	**0.005** 0.610	836.236
Dominant	AG + GG vs. AA	1.674 (1.170–2.394)	**0.0** **05**	834.284
Recessive	GG vs. AG + AA	1.247 (0.309–5.037)	0.756	841.985
Overdominant	AG vs. AA + GG	1.675 (1.165–2.408)	**0.005**	834.486
Additive	G	1.572 (1.129–2.188)	**0.007**	835.030

OR—odds ratio; CI—confidence interval; *p*-value—significance level; AIC—Akaike information criteria.

**Table 4 medicina-58-00653-t004:** Genotype and allele distribution of *SIRT1* rs3818292, rs3758391, and rs7895833 in periodontitis and control groups between males.

Genotype/Allele	Periodontitis *n* = 85 *n* (%)	Control Group *n* = 250 *n* (%)	*p*-Value
**Males**			
**rs3818292**			
AA	66 (77.6) ^1^	222 (88.8) ^1^	**0.024**
AG	19 (22.4) ^2^	27 (10.8) ^2^	
GG	0 (0)	1 (0.4)	
Total	85 (100.0)	250 (100.0)	
A	151 (88.8)	471 (94.2)	**0.019**
G	19 (11.2)	29 (5.8)	
**rs3758391**			
CC	43 (50.6)	144 (57.6)	0.631
CT	36 (42.4)	96 (38.4)	
TT	6 (7.1)	10 (4.0)	
Total	85 (100.0)	250 (100.0)	
C	122 (71.8)	384 (76.8)	0.187
T	48 (28.2)	116 (23.2)	
**rs7895833**			
AA	51 (60.0) ^3^	188 (75.2) ^3^	**0.027**
AG	32 (37.6) ^4^	59 (23.6) ^4^	
GG	2 (2.4)	3 (1.2)	
Total	85 (100.0)	250 (100.0)	
A	134 (78.8)	435 (87.0)	**0.010**
G	36 (21.2)	65 (13.0)	

*p*-value—significance level; ^1^ (AA vs. AG + GG) *p* = 0.010; ^2^ (AG vs. AA + GG) *p* = 0.008; ^3^ (AA vs. AG + GG) *p* = 0.007; ^4^ (AG vs. AA + GG) *p* = 0.012.

**Table 5 medicina-58-00653-t005:** Genotype and allele distribution of *SIRT1* rs3818292, rs3758391, and rs7895833 in periodontitis and control groups between females.

Genotype/Allele	Periodontitis *n* = 116 *n* (%)	Control Group *n* = 250 *n* (%)	*p*-Value
**Females**			
**rs3818292**			
AA	96 (82.8)	224 (89.6)	0.124
AG	18 (15.5)	25 (10.0)	
GG	2 (1.7)	1 (0.4)	
Total	116 (100.0)	250 (100.0)	
A	210 (90.5)	473 (94.6)	**0.039**
G	22 (9.5)	27 (5.4)	
**rs3758391**			
CC	60 (51.7)	144 (57.6)	0.570
CT	50 (43.1)	94 (37.6)	
TT	6 (5.2)	12 (4.8)	
Total	116 (100.0)	250 (100.0)	
C	170 (73.3)	382 (76.4)	0.361
T	62 (26.7)	118 (23.6)	
**rs7895833**			
AA	82 (70.7)	195 (78.0)	0.267
AG	33 (28.4)	52 (20.8)	
GG	1 (0.9)	3 (1.2)	
Total	116 (100.0)	250 (100.0)	
A	197 (84.9)	442 (88.4)	0.188
G	35 (15.1)	58 (11.6)	

*p*-value—significance level.

**Table 6 medicina-58-00653-t006:** Genotype and allele distribution of *SIRT1* rs3818292, rs3758391, and rs7895833 in periodontitis and control groups within males.

Model	Genotype/Allele	OR (95% CI)	*p*-Value	AIC
**Males**				
***SIRT1* rs3818292**				
Co-dominant	AG vs. AA GG vs. AA	2.367 (1.238–4.525) -	**0.009** -	376.41
Dominant	AG + GG vs. AA	2.282 (1.199–4.347)	**0.012**	375.46
Recessive	GG vs. AG + AA	-	-	-
Overdominant	AG vs. AA + GG	2.378 (1.244–4.545)	**0.009**	374.93
Additive	G	2.110 (1.129–3.945)	**0.019**	376.24
***SIRT1* rs3758391**				
Co-dominant	CT vs. CC TT vs. CC	1.256 (0.752–2.097) 2.009 (0.691–5.846)	0.384 0.200	381.53
Dominant	CT + TT vs. CC	1.327 (0.810–2.174)	0.261	380.23
Recessive	TT vs. CT + CC	1.823 (0.642–5.175)	0.259	380.28
Overdominant	CT vs. CC + TT	1.179 (0.715–1.943)	0.520	381.07
Additive	T	1.329 (0.882–2.002)	0.174	379.66
***SIRT1* rs7895833**				
Co-dominant	AG vs. AA GG vs. AA	1.999 (1.177–3.397) 2.458 (0.400–15.103)	**0.010** 0.332	376.55
Dominant	AG + GG vs. AA	2.022 (1.201–3.401)	**0.008**	374.60
Recessive	GG vs. AG + AA	1.984 (0.326–12.079)	0.457	380.97
Overdominant	AG vs. AA + GG	1.955 (1.154–3.311)	**0.013**	375.42
Additive	G	1.888 (1.173–3.038)	**0.009**	374.78

OR—odds ratio; CI—confidence interval; *p*-value—significance level; AIC—Akaike information criteria.

**Table 7 medicina-58-00653-t007:** Genotype and allele distribution of *SIRT1* rs3818292, rs3758391, and rs7895833 in periodontitis and control groups between patients older than 60 years old.

Genotype/Allele	Periodontitis *n* = 151 *n* (%)	Control Group *n* = 368 *n* (%)	*p*-Value
**rs3818292**			
AA	118 (78.1) ^1^	329 (89.4) ^1^	**0.001**
AG	31 (20.5) ^2^	39 (10.6) ^2^	
GG	2 (1.3)	0 (0)	
Total	151 (100.0)	368 (100.0)	
A	267 (88.4)	697 (94.7)	**0.001**
G	35 (11.6)	39 (5.2)	
**rs3758391**			
CC	74 (49.0)	209 (56.8)	0.187
CT	67 (44.4)	144 (39.1)	
TT	10 (6.6)	15 (4.1)	
Total	151 (100.0)	368 (100.0)	
C	215 (71.2)	562 (76.4)	0.081
T	87 (28.8)	174 (23.6)	
**rs7895833**			
AA	97 (64.2) ^3^	280 (76.1) ^3^	**0.022**
AG	51 (33.8) ^4^	84 (22.8) ^4^	
GG	3 (2.0)	4 (1.1)	
Total	151 (100.0)	368 (100.0)	
A	245 (81.1)	644 (87.5)	**0.008**
G	57 (18.9)	92 (12.5)	

*p*-value—significance level; ^1^ (AA vs. AG + GG) *p* = 0.001; ^2^ (AG vs. AA + GG) *p* = 0.003; ^3^ (AA vs. AG + GG) *p* = 0.006; ^4^ (AG vs. AA + GG) *p* = 0.001.

**Table 8 medicina-58-00653-t008:** Binomial logistic regression analysis of periodontitis and control groups in the older than 60-year-old patients’ group.

Model	Genotype/Allele	OR (95% CI)	*p*-Value	AIC
**Periodontitis**				
***SIRT1* rs3818292**				
Co-dominant	AG vs. AA GG vs. AA	2.216 (1.322–3.714) 4.5 × 10^9^ (0.000)	**0.003** 1.0	616.12
Dominant	AG + GG vs. AA	2.359 (1.418–3.925)	**0.001**	617.31
Recessive	GG vs. AG + AA	4.0 × 10^9^ (0.000)	1.0	622.95
Overdominant	AG vs. AA + GG	2.179 (1.301–3.650)	**0.003**	619.43
Additive	G	2.408 (1.474–3.933)	**0.001**	615.81
***SIRT1* rs3758391**				
Co-dominant	CT vs. CC TT vs. CC	1.314 (0.887–1.946) 1.883 (0.810–4.374)	0.173 0.141	626.63
Dominant	CT + TT vs. CC	1.368 (0.935–2.000)	0.106	625.30
Recessive	TT vs. CT + CC	1.669 (0.732–3.803)	0.223	626.48
Overdominant	CT vs. CC + TT	1.241 (0.846–1.820)	0.270	626.70
Additive	T	1.340 (0.976–1.840	0.070	624.65
***SIRT1* rs7895833**				
Co-dominant	AG vs. AA GG vs. AA	1.753 (1.154–2.661) 2.165 (0.476–9.846)	**0.008** 0.318	622.50
Dominant	AG + GG vs. AA	1.771 (1.176–2.669)	**0.006**	620.57
Recessive	GG vs. AG + AA	1.845 (0.408–8.342)	0.427	627.30
Overdominant	AG vs. AA + GG	1.724 (1.138–2.614)	**0.010**	621.44
Additive	G	1.688 (1.157–2.463	**0.007**	620.67

OR—odds ratio; CI—confidence interval; *p*-value—significance level; AIC—Akaike information criteria.

## Data Availability

All data can be observed in this manuscript.

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
