# Peer review of "Value of Serum Sirtuin-1 (SIRT1) Levels and SIRT1 Gene Variants in Periodontitis Patients"

_medicina, 2022, doi:10.3390/medicina58050653_

Round 1

Reviewer 1 Report

The contents of the manuscript were useful and informative. Please find my comment in the attachment

Reviewer 2 Report

  1. In Material and method section please explain about the sampling method of this study
  2. one of the exclusion criteria of this study that tooth loss wasnot due to periodontal disease  but to other causes --> please add information  : how can you  get the information about the cause of tooth loss  since the tooth already lossin the time of study
  3. Please  explain  about the design of your study , whether it is a case control study since you  choose sample in 2 groups  cases and control ? but I didnt find any information in the study about this
  4. Is there any other variables included in your study such as oral hygienne  ofthe subjects ? since those variables is the confounding factors that can affect the result of your study.Please add information about this.

Reviewer 3 Report

sample size calculation has not been performed. I can be argued that the sample is not big enough to draw definitive conclusions.
